# Analysis of Sulfated Glycosaminoglycans in ECM Scaffolds for Tissue Engineering Applications: Modified Alcian Blue Method Development and Validation

**DOI:** 10.3390/jfb10020019

**Published:** 2019-04-30

**Authors:** Tuyajargal Iimaa, Yasuhiro Ikegami, Ronald Bual, Nana Shirakigawa, Hiroyuki Ijima

**Affiliations:** 1Department of Chemical Engineering, Graduate School of Engineering, Kyushu University, 744 Motooka, Nishi-Ku, Fukuoka 819-0395, Japan; tuyaiimaa@gmail.com (T.I.); y-ikegami@kyudai.jp (Y.I.); ronald.bual@g.msuiit.edu.ph (R.B.); shirakigawa@chem-eng.kyushu-u.ac.jp (N.S.); 2Department of Biochemistry and Laboratory Medicine, School of Biomedicine, Mongolian National University of Medical Sciences, Ulaanbaatar 14210, Mongolia; 3Department of Chemical Engineering & Technology, College of Engineering, Mindanao State University-Iligan Insititute of Technology, Iligan 9200, Philippines

**Keywords:** sulfated glycosaminoglycans, heparin, alcian blue method, method validation, effect of protein, liver-derived extracellular matrix, tissue engineering

## Abstract

Accurate determination of the amount of glycosaminoglycans (GAGs) in a complex mixture of extracellular matrix (ECM) is important for tissue morphogenesis and homeostasis. The aim of the present study was to investigate an accurate, simple and sensitive alcian blue (AB) method for quantifying heparin in biological samples. A method for analyzing heparin was developed and parameters such as volume, precipitation time, solvent component, and solubility time were evaluated. The AB dye and heparin samples were allowed to react at 4 ℃ for 24 h. The heparin-AB complex was dissolved in 25 N NaOH and 2-Aminoethanol (1:24 *v*/*v*). The optical density of the solution was analyzed by UV-Vis spectrometry at 620 nm. The modified AB method was validated in accordance with U.S. Food and Drug Administration guidelines. The limit of detection was found to be 2.95 µg/mL. Intraday and interday precision ranged between 2.14–4.83% and 3.16–7.02% (n = 9), respectively. Overall recovery for three concentration levels varied between 97 ± 3.5%, confirming good accuracy. In addition, this study has discovered the interdisciplinary nature of protein detection using the AB method. The basis for this investigation was that the fibrous protein inhibits heparin-AB complex whereas globular protein does not. Further, we measured the content of sulfated GAGs (sGAGs; expressed as heparin equivalent) in the ECM of decellularized porcine liver. In conclusion, the AB method may be used for the quantitative analysis of heparin in ECM scaffolds for tissue engineering applications.

## 1. Introduction

Tissue engineering involves the use of a combination of cells, biocompatible scaffold materials, and suitable biochemical factors (e.g., growth factors) to create tissue-like structures and support for regeneration. Scaffolds are primarily composed of proteins and polysaccharides such as proteoglycans (PGs) and glycosaminoglycans (GAGs) that display structural and chemical properties of the native extracellular matrix (ECM) and influence biological activity to promote and regulate the regeneration of tissues and organs [1,2,3,4]. A study conducted in 1989 demonstrated the potential of PG-coated dishes as a scaffold to enable the self-assembly of primary hepatocytes into multicellular spheroids, which exhibited enhanced liver-specific function. In recent years, there has been heightened interest in using decellularized tissue matrices obtained from processing discarded donor tissue. For example, H. Ijima et al. reported that the solubilized ECM derived from decellularized liver (L-ECM) has the potential to act as an effective scaffold material in the field of regenerative medicine [1,5,6]. 

PGs and proteins are the two main classes of macromolecules present in the ECM; PGs are major constituents of connective tissues composing blood vessels, cartilage, skin, and neural tissues. Due to their ability to interact with other molecules like cytokines and growth factor receptors, they play a crucial role in cell signaling [7,8]. PGs are bio-macromolecules composed of sulfated GAG (sGAG) chains covalently linked to a protein core. It should be emphasized that sGAG chains form porous hydrated gels and fill most of the extracellular space. The biological functions of PGs depend on the interaction of the sGAG chains with different protein ligands [9]. 

One of the most common sGAGs is heparin, which consists of repeating units of 1,4-linked pyranosyluronic acid and 2-amino-2-deoxyglucopyranose (glucosamine) residues. Heparin has the highest negative charge density of any known biological macromolecule, where at least 70–80% of heparin is composed of a trisulfated disaccharide, and the molecular weight distribution of heparin corresponds to its polydispersity. This structural variability makes heparin an extremely challenging molecule to characterize, more sulfated, and is thus more charged than other sGAGs. Furthermore, it is one of the well-known molecules used for tissue regeneration applications [10,11,12]. sGAGs play an important role in chemical signaling between cells by binding to and regulating the activity of various secreted signaling molecules and proteins [9,12,13].

High performance liquid chromatography [14], nuclear magnetic resonance [15], mass spectrometry (MS) [16], liquid chromatography-MS [17], high-performance anion-exchange chromatography [18], carbazole assay [19], toluidine blue [20], methylene blue [21], and AB assays [22] have been developed to quantify and analyze sGAGs such as heparin in various samples. However, these techniques are time-consuming, sensitive to contamination, costly, and involve complicated procedures. Therefore, there is a need to develop a fast, reliable, and simple method for quantifying heparin. Heparin is widely used as a polyelectrolyte in a multilayer built-up ECM and also for the ECM component of hydrogel materials in tissue engineering applications. Physicochemical studies identify that the thermodynamics and kinetics of protein–heparin binding show a considerable increase in sensitivity over other sGAGs [3,4,11].

More general methods have been established; these quantify heparin by measuring the negatively charged side chains. AB is a well-established assay, which is advantageous over other methods by being reproducible and simple to perform in general lab settings. In numerous studies, AB is commonly used in combination with sGAGs for various methods of quantification and qualification [23]. AB is a polyvalent basic dye with a hydrophobic copper core that interacts electrostatically with tissue-derived polyanionic molecules, such as heparin. The binding of AB to heparin chains is proportional to the number of negative charges present on the heparin chain [24]. In addition, sGAGs are covalently bonded to core proteins residing in the ECM as well as proteins containing a number of negatively charged carboxyl groups and a hydrophobic domain. These carboxyl groups in a protein can be eliminated using the AB method at a low pH, which might hinder the electrostatic interaction between sGAG and AB [23,24]. Clearly, the most prominent interaction type between heparin and a protein is the ionic interaction [12]. Moreover, Alberto L. et al. suggested that the proteins can cause interference with the quantification of carbohydrate [25]. It is very interesting and important to elucidate how heparin-AB complex is affected by the protein composition. 

Our study has focused on the development of an accurate, simple, and reliable AB method to quantify heparin in a sample. Furthermore, the quantitative AB method was validated in accordance with U.S. FDA and ICH guidelines in terms of linearity, limit of detection, limit of quantification, precision, and accuracy [26,27]. This study has also discovered the effect of protein in the AB method. The basis for this investigation is indicating that fibrous protein inhibits the formation of a heparin-AB complex, whereas globular protein does not. This effect may be due to the molecular structure of the proteins. After being developed, the AB method has been successfully applied for the determination of the sGAG amount (expressed as heparin equivalent) in ECM scaffolds from the porcine liver. This AB method will prove to be very meaningful and effective for quantifying the heparin amount in ECM scaffolds for tissue engineering applications.

## 2. Results

### 2.1. Optimization of the AB Method Conditions

#### 2.1.1. The Volume Ratio of the Sample and AB Dye

In this study, two different volumes of the heparin standard (25 µL and/or 40 µL) were used in triplicate (n = 3) to determine the desired sample volume. A calibration curve was generated by plotting a graph of optical density versus heparin concentration. It is evident that the absorbance or optical density (OD) of a light-absorbing material determined according to the Beer-Lambert’s law is directly proportional to the solute concentration in the solution [28]. As seen in Figure 1, the optical density of 40 µL of the sample and AB dye solution was higher than that of 25 µL of the sample, and the correlation coefficient for 25 µL of sample and 40 µL of sample was calculated as 0.996 and 0.998, respectively. Test results show that 25 µL of sample and 25 µL of AB dye solution had comparatively lower optical density. Therefore, the volume of 40 µL was selected for further experiments. 

#### 2.1.2. Precipitation Time

Figure 2 shows the correlation between precipitation time and electrostatic attraction between heparin-AB for 24 h (short), 48 h (medium), and 72 h (long). During the experimental period, the optical density of the heparin-AB complex was found to be stable for almost 72 h. It is important to note that our main purpose was to develop a time-efficient method; therefore, we have finalized the precipitation time of 24 h for the optimized AB method. 

#### 2.1.3. Solvent Component

The solvent component was optimized in terms of solvent concentration and compounds’ ratio. Thus, various types of solvent components with NaOH and 2-Aminoethanol were tested to select an appropriate solvent component (Table 1). We measured the optical density of a sample (OD.1), then diluted it 2.5 times using the solvent component and measured the optical density (OD.2) again. An adequate solvent component was selected according to the result of their dilution ratio. A suitable solvent component could reach a dilution ratio of 2.5. 

The data presented in Figure 3 clearly observed that the dilution ratios for the first and second solvents were insufficient while the dilution ratio of the third solvent was higher than that of other solvents. The solvent component consisting of 25 N NaOH and 2-Aminoethanol (1:24, *v*/*v*) was determined to be the appropriate solvent component and thus selected for the AB method. 

#### 2.1.4. Solubility Time

In addition to the solvent component study, we assessed the solubility time by dissolving the precipitation formed upon the heparin-AB interaction. The solubility was monitored at different time intervals: 0 min, 15 min, and 30 min after adding 1 mL of solvent component (Figure 4a). The dilution process has been determined in a similar way as described previously for solvent component studies. Briefly, after reading OD.1, 600 µL of the solvent component was added to the sample tubes and then incubated for different periods of time to completely dissolve the heparin-AB complex. The samples were centrifuged and finally, the OD.2 was measured at 620 nm again to indicate the dilution ratio. The difference in optical density at three different time points is minimal whereas the dilution ratios for 0 (zero solubility time), 15 min and 30 min were 2.2, 2.4 and 2.5, respectively, as shown in Figure 4b.

It is interesting to note that the dilution ratio for zero solubility time was lower than that for other cases while the dilution ratios at 15 min and 30 min have given the best results. The solubility time for 30 min was chosen as this result is close to the theoretical value of dilution ratio. Finally, we reported the general description of the modified AB method in Figure 5.

### 2.2. Method Validation

The modified AB method was validated in terms of linearity, Limit of detection (LOD) and quantification (LOQ), accuracy, intraday precision (repeatability) and interday precision.

#### 2.2.1. Linearity

The linearity of an analytical procedure is the capacity (within a given range) to obtain test results that are directly proportional to the concentration of an analyte in samples [27]. The response of the heparin standard was found to be linear in the investigated concentration range and the regression equation was y = 0.0013x + 0.003, as shown in Table 2. The results showed good linearity for heparin, with a correlation coefficient (r^2^) of approximately 0.996 in all cases (Figure 1 and Figure 8b). It can be seen from test results that the calibration curves for heparin were linear within a range of 12.5–400 µg/mL.

#### 2.2.2. Limit of Detection (LOD) and Quantification (LOQ)

In the present study, LOD was found to be 2.95 µg/mL, while the LOQ value for heparin was found to be 9.82 µg/mL. Here, the LOD value indicates the high sensitivity of the proposed method. Table 3 summarizes the main results from method validation. 

#### 2.2.3. Accuracy

Accuracy was ascertained by standard samples at different concentrations (25 µg/mL, 50 µg/mL and 100 µg/mL) of heparin. Recovery is often used as a measure of accuracy. This number was determined by calculating the percent mean recovery of the experimentally determined concentration compared to the nominal one [26,27]. The results were expressed as percentage recovery, which was in the range between 97.5 ± 3.5%, and indicated a high degree of accuracy and absence of interference. The results of the recovery study are shown in Table 4.

#### 2.2.4. Precision

The precision of the method was evaluated by determining the relative error (RE, %), and coefficient of variation (CV) in the intraday and interday variation. The CV (%) was calculated by assessing the standard deviation and mean value [26]. The intraday precision was determined by duplicate assays at different spiking levels (25 µg/mL, 50 µg/mL and 100 µg/mL) of heparin standards at the same time. Herein, the CV values determined for an intraday variation were 2.14–4.80%. In addition, the developed method was found to be precise as the CV values determined for interday variation under the same conditions for 3 days were between 3.16% and 7.02%. Table 5 represents the intraday and interday precisions from this study, where CV did not exceed 10% in either case. We also calculated the mean value of precision, SD and RE in all cases. The RE of 25 µg/mL, 50 µg/mL and 100 µg/mL sample ranged from 0.34 to 1.19 in the experiment. These statistics were necessary to inspect the variation in datasets. 

### 2.3. The Effect of Protein Structure on the Modified AB Method

The results of heparin alone, heparin with gelatin and heparin with bovine serum albumin (BSA) samples at different concentrations are shown in Figure 6. As mentioned above, we used also the fibrous protein as gelatin while the globular protein as BSA in this study. In all cases, the optical density of the sample that has different heparin concentration was determined simultaneously. The optical density of heparin with BSA samples had identical values for different concentrations of heparin. This indicates that globular protein does not affect the heparin-AB interaction, while marginal decreases were detected in heparin with gelatin samples when compared to other samples. This result suggests that the quantification reaction between heparin and AB was subtly influenced by the fibrous protein as gelatin.

### 2.4. Quantification of sGAGs in the L-ECM

An optimized AB method was used to accurately determine the amount of sGAGs in native tissue L-ECM derived from decellularized porcine liver. The experimental design for the preparation of native L-ECM tissue from the decellularized porcine liver scaffold is shown in Figure 7. 

According to the assay protocol, the collagen quantitation kit was used to determine the concentration of fibrous proteins. Then, the optical density of fibrous proteins in the L-ECM was calculated from the calibration curve of heparin with gelatin samples (Figure 8a). Our result showed that fibrous proteins reduced the optical density to approximately 0.012. Then, the amount of heparin was calculated accurately from the calibration curve of heparin within a concentration range of 12.5–400 µg/mL. The dose-response lines of the heparin with the serial concentration are presented in Figure 8b. The heparin amount in L-ECM was 40.65 µg/mL, expressed as heparin equivalent. 

## 3. Discussion

### 3.1. Assay Optimization

AB has been one of the most widely used cationic dyes for acidic polysaccharides such as sGAGs since 1960. The AB method is based on the formation of a deep blue complex between negatively charged sGAGs and positively charged AB that precipitates out. It was reported that the heparin and AB precipitate is insoluble in water as the ionic bonding between sGAG and AB is a stable interaction [22,23]. As a tetravalent cation—with a hydrophobic core with positive charges attached strongly to heparin—the ionic strength and presence of other interactions affect the precipitate in solution [29]. 

In this study, the volume of sample and AB dye was tested 25 µL and 40 µL, and the regression coefficients were calculated. Figure 1 shows that the optical density of the 40 µL sample and AB was higher than the optical density of 25 µL sample and AB. 

As noted above, heparin is the most negatively charged biomolecule, and the interaction of heparin with AB dye produces a blue precipitate that is shown in Figure 4a. The precipitation time conditions (Figure 2) for 72 hours can appear that the time of heparin-AB interaction. Data showed that the extension of time for the precipitation process was unnecessary; a time of 24 h was adequate for the ionic interaction to take place between heparin and AB. 

In addition, solvent component and solubility time were established using three different cases. The difference of the optical density was determined by optical density of the sample and diluted sample. Results are shown in Table 1, Figure 3 and Figure 4b. Our results of experiments showed that the third solvent component can dissolve the heparin-AB complex for 30 min. 

Finally, we have been established the AB method where: 40 µL of sample and 40 µL of AB reacted for 24 h in the dark at 4 ℃; the solvent component was dissolved using 25 N NaOH and 2-Aminoethanol, then incubated for 30 min; lastly, the OD was measured at 620 nm to determine the amount of heparin in a sample. 

In contrast to the AB method, other methods are unnecessarily time-consuming or expensive [14,15,16,17,18,19]; toluidine, methylene blue, and other staining method have not been fully explored for routine heparin quantification [20,21]. Our newly developed AB method has the advantage of considerably lower consumption of reagents, which are commonly available and easy to prepare chemicals. Therefore, it is cost-effective and requires less analysis time as well. In brief, the modified AB method is simple and can be readily implemented with the instruments available in a general laboratory setting.

### 3.2. Method Validation

Heparin quantification in biological samples and pharmaceutical products is important for the verification of their quality and is mandatory by many regulatory agencies such as the U.S Food and Drug Administration (FDA) [26]. For this purpose, we validated the modified AB method to quantify heparin. Performance characteristics such as regression coefficient, limit of detection (LOD), limit of quantification (LOQ), intra-and-interday precisions, and accuracy were considered for the validation of the new method. 

The detectable range of the AB method is established between the highest and lowest concentrations with satisfactory accuracy and precision. [22,23]. Under the working conditions, the linear range 12.5–400 µg/mL of heparin was obtained with a six-point standard curve. The results are shown in Table 2 and Table 3. In addition, LOD was the lowest concentration of the analyte that can be detected by a given method; LOQ was the smallest amount of heparin that can be quantitatively measured in a sample with acceptable accuracy and precision. [27]. The value of LOD was 2.95 µg/mL, and LOQ was 9.82 µg/mL (Table 3), thus indicating low LOD and LOQ variation.

Table 4 shows results from accuracy in each analyzed sample. To determine the accuracy of the method, the recovery was found at different concentrations of heparin. The accuracy test for these concentration levels ranged between 94.97 and 100.97%, confirming the reliability of the AB method for determination of the amount of heparin. 

The precision of the modified method was calculated according to the International Conference on Harmonization (ICH) guidelines. When the analytical method is repeatedly applied to multiple aliquots of the sample, the precision is the homogeneity of values within a series of individual measurements of an analyte [27]. For the experimental conditions, the CV precision percent of the inter-day (repeatability) ranged 2.14–4.80%. Intra-day (intermedia) variation was calculated by the comparison of sGAGs concentrations obtained from the sample (n = 27) performed in three separate assays and the CV found from 3.16 to 7.02%. For both precision assays in inter-and-intra-days were CV lower than the acceptance criteria or 5% and 10%, respectively. Furthermore, RE was around ±1.19 (Figure 5). 

Results from the AB method can be used to evaluate the quality and consistency of analytical procedures. Based on these evaluations, we can claim that the developed AB method has been well-validated. 

### 3.3. Protein Effect

Sulfated polysaccharides are capable of very specific interactions with proteins under physiological conditions [29,30,31]. Some researchers have also considered that the presence of the protein may interfere with carbohydrate estimation [25]. Depending upon the chemical structure, proteins are classified as either fibrous or globular [32]. Therefore, the effect of fibrous and globular proteins on the heparin-AB interaction was determined separately. In addition, some studies mentioned that heparin structure has the capability of interacting with more than one protein and activity should be viewed at the level of their interactions with multiple proteins [11,12]. sGAGs are capable of specific D-periodic binding with collagen fibrils even in the absence of protein cores, an interaction too specific to be due to only the collagen being fibrous [33]. Therefore, it remains an open question as to whether all heparin sequences interact with proteins to some degree and whether synergy effects are more relevant [25]. 

We tested further to determine that (Figure 6). An evaluation of the spectral properties of the samples showed that it has different spectral properties to gelatin with a heparin sample. Thus, compared with heparin, the optical density was decreased (p < 0.05). On the other hand, the heparin-AB interaction was reduced by gelatin. These results confirmed that fibrous protein was affected the optical density of the heparin sample. However, this negative aspect of heparin with gelatin can be explained by fibrous protein behavior. Consequently, fibrous proteins only inhibit the heparin-AB interaction. Globular protein did not interfere in this study. Physically, fibrous proteins are less sensitive whereas globular proteins are more sensitive to changes in pH, temperature, etc. In terms of amino acid sequence, fibrous proteins contain highly repetitive sequences while globular proteins have irregular sequences. Protein behavior can be predicted by the amino acid sequence [34]. It is interesting to investigate the understandings of the chemical properties of the fibrous protein for this method of quantifying heparin. 

### 3.4. Method Application

ECM is a non-cellular 3D macromolecular network that is composed of collagen, elastin, fibronectin, laminins, proteoglycans/glycosaminoglycans, and several other glycoproteins. ECM scaffolds have been developed for tissue engineering applications in recent years [2]. Theoretically, hundreds of proteins interact with sGAGs, which are directly involved in various signaling pathways. In addition, sGAG biosynthesis is regulated by a variety of chemokines, cytokines, and growth factors that can enhance or inhibit cell signaling activity [9,12]. In both in vitro and in vivo studies, researchers have been able to detect the exact amount of sGAG.

Firstly, we prepared L-ECM as described above. The AB method designed in this paper has been used for the analysis of heparin amount; triplicate measures were taken. The optical density of L-ECM was compared to a standard based on heparin, then total sGAG content was expressed as µg/mL of heparin. In such sense, the use of the AB method allows heparin determination in the sample quickly and easily. 

In summary, the AB method, monitored by spectrometry, has been developed for the quantification of heparin in samples. The modified AB method was validated in terms of parameters, including a determination coefficient, accuracy, coefficient of variation, quantification limit, and detection limit. Interestingly, we found that fibrous proteins inhibit the formation of the heparin-AB complex, whereas globular proteins do not. The difference might be due to the type and chemical structure of the proteins. Finally, we took the ECM derived from the porcine liver and determined the amount of sGAGs it contained, considering the presence of proteins in the L-ECM, and expressing the results in terms of heparin equivalents against a heparin standard curve. As ECM sGAGs have substantially different structures to heparin, the value obtained is likely to be inaccurate in absolute terms. Additionally, our finding suggested that this modified AB method may be applied to artificial, heparin-containing scaffolds and other wide-ranging applications in the tissue engineering field.

## 4. Materials and Methods

### 4.1. Chemicals and Reagents

The materials and reagents in this study were obtained from the following sources: heparin sodium (1,000,000 units), albumin from bovine serum (BSA, Cohn Fraction V, pH 7.0), and 2-Aminoethanol were purchased from Wako Pure Chemicals (Osaka, Japan). Gelatin from porcine skin (gel strength 300 Type A), Triton X-100, pepsin from porcine gastric mucosa (lyophilized powder, ≥2500 units/mg protein), and Alcian blue 8GX were purchased from Sigma Aldrich Ltd (St. Louis, MO, USA). Collagen type I-C was acquired from Nitta Gelatin Inc. (Osaka, Japan) and collagen quantitation assay kit was acquired from Cosmo Bio (Tokyo, Japan). All other chemicals were of analytical grade. The healthy porcine liver was harvested from adult pigs, Fukuokashokunikuhanbai Co. Ltd. (Fukuoka, Japan).

### 4.2. Preparation of Solutions Used in the Method Developing Process

In this study, 10 mg heparin per 1 mL of ultrapure water was used as a standard solution from which serial dilutions of heparin standards were prepared in the range of 12.5–400 µg/mL.

Thirty milligrams of both gelatin and bovine serum albumin (BSA) were weighed and dissolved in 1 mL of ultrapure water, and this was further diluted to 15 mg/mL, 10 mg/mL and 5 mg/mL. Gelatin, a fibrous protein, was completely dissolved at 37 ℃ in 30 min, while BSA, a globular protein, was dissolved in 10 min.

For 0.1% (*w*/*v*) Alcian blue 8GX solution, 10 mg of AB dissolved in 1 mL of 1 N hydrochloric acid (HCl). 

### 4.3. Decellularization and Solubility of the Porcine Liver

The liver was depleted of blood in a fresh state with calcium and magnesium-free phosphate-buffered saline (CMF-PBS) and cryopreserved at −80 °C until use. A porcine liver was sliced to 1 cm × 1 cm × 2 mm pieces using a mandolin-style slicer and decellularized using 1% Triton X-100 in CMF-PBS at 4 °C for 3 days to remove the cellular components. The solution was changed each day for 3 days under constant stirring to maintain the efficiency of decellularization. The resulting decellularized liver was immersed in CMF-PBS at 4 °C for 4 days to remove the detergent. Dialysis was performed to remove salts and impurities using the Spectra/Por 6 dialysis membrane (MCWO: 1000, Spectrum Laboratories, Inc., Milpitas, CA, USA). Finally, lyophilization was done for 24 h to remove moisture. The dried L-ECM was powdered in a mill mixer and 10 mg of dried L-ECM powder was solubilized in 1 mL of pepsin solution (1 g/mL in 0.01 N HCl) and then constantly stirred for 3 days at 4 °C. After that, the ECM solution was centrifuged to remove the supernatant and dialyzed with 1 L ultrapure water for 24 h. The resulting L-ECM pre-gel (pH = 3.0–4.0) was stored at 4 °C until further analysis. All animal experiments were performed in accordance with the guidelines of the Ethics Committee on Animal Experiments and accepted by Kyushu University (A27-326-0, 19 February 2016).

### 4.4. The Modified AB Method

Sample and 10 mg/mL of AB dye were mixed with at a ratio of 1:1. Sample tubes were then placed in the dark and incubated for 24 h to facilitate the formation of the precipitate. The remaining dye was removed by centrifugation at 10,000 rpm for 10 min at 4 °C. The precipitate was resuspended using 1 N hydrochloric acid (HCl) and centrifuged again at 10,000 rpm for 10 min at 4 °C to remove the supernatant. Meanwhile, a solvent component was prepared by dissolving 25 N sodium hydroxide (NaOH) solution in 2-Aminoethanol (1:24 *v*/*v*) until the mixture became homogeneous. Subsequently, 1 mL of the solvent component was added to the precipitate, mixed using a pipette, vortexed, then placed in the dark for 30 min and finally centrifuged. All the processes were repeated thrice to completely dissolve the precipitate. After the third centrifugation, 200 µL volume of each sample was transferred in duplicates into a 96-well plate, while the solvent solution was used as a blank. Optical density (OD) was measured at 620 nm within 5 min after the microplate procedure. OD measurement was performed on a Microplate Photometer from Thermo Scientific (MA, USA).

### 4.5. Optimization of the Modified AB Method

Based on the previous study [20], several tests were conducted to optimize the AB method. For instance, 25 µL and 40 µL volume of each sample was mixed with AB dye. The crystallization time was tested at 24 h, 48 h, and 72 h. Meanwhile, a number of solvent components were prepared with NaOH and 2-Aminoethanol and lastly, solubility time was varied for 0 min, 15 min, and 30 min. While deciding on the solvent component and solubility time, a dilution method for heparin absorption was followed. The dilution ratio of 2.5 was a good indicator in all cases. 

The microplate procedure was carried out as described earlier and the optical density (OD.1) of 400 µL of sample was measured. Thereafter, 600 µL of the solvent component was added to the sample. This solution was then mixed using a pipette, vortexed, then incubated for 30 min in the dark and finally centrifuged at 10,000 rpm for 10 min at 4 °C. A volume of 200 µL of each diluted sample was placed in duplicates into a 96-well plate and the optical density (OD.2) was measured at 620 nm again. Alternatively, the value of OD.2 was equal to the dilution of OD.1 by 2.5 times.

### 4.6. Validation of the Modified AB Method

The linearity, regression coefficient, y-intercept, slope of the regression line, limit of detection (LOD), limit of quantification (LOQ), intra-and-inter-day precisions and accuracy were analyzed. 

Linearity was performed using a calibration curve corresponding to heparin concentration of 12.5–400 µg/mL. Triplicates prepared separately at each concentration were analyzed. 

LOQ is the minimum quantifiable concentration that was calculated using the standard formula:(1)3·σ/S,
while LOQ was calculated as:(2)10·σ/S
where σ is the standard deviation of the response, and S is the slope of the linear regression [25,26]. 

Accuracy was calculated as the percentage recovery based on the given formula:(3)$$\mathrm{Accuracy}\;\left(\%\right)=\;\frac{\mathrm{spiked}\;\mathrm{concentration}/\mathrm{nominal}\;\mathrm{concentration}}{100}$$

The intraday precision, interday precision, and accuracy were estimated by analyzing nine replicates at three different heparin concentrations of 25 µg/mL, 50 µg/mL and 100 µg/mL. The precision was expressed as the coefficient of variation (CV) and relative error (RE) of the replicate measurements
(4)$$\mathrm{CV}\;=\;\frac{\mathrm{SD}}{\mathrm{mean}}\times 100\%,$$
(5)$$\mathrm{RE}=\;\frac{\mathrm{nominal}\;\mathrm{concentration}-\mathrm{spiked}\;\mathrm{concentration}}{\mathrm{spiked}\;\mathrm{concentration}}\times 100\%$$

The intraday precision was illustrated by nine replicates of heparin samples while the interday precision was determined by these heparin samples on three consecutive days. The acceptance criteria of the data are clearly defined by the FDA guidelines for bioanalytical method validation [24]. 

### 4.7. Determination of the Effect of Protein Structure on the Modified AB Method

In order to determine the effect of protein structure on the heparin quantification, fibrous and globular proteins were selected. Different concentrations of heparin (100 µg/mL, 200 µg/mL and 400 µg/mL), gelatin (5 mg/mL), and BSA (5 mg/mL) were prepared. The effect of protein structure on the AB method was determined by the optical density of these samples as mentioned above.

### 4.8. The Amount of sGAG in Solubilized L-ECM

To investigate the efficacy of the modified AB method, the amount of sGAGs in solubilized L-ECM was determined with reference to the fibrous protein content. Firstly, the amount of fibrous proteins in the L-ECM scaffold was assessed by a collagen quantification assay kit that uses a fluorogenic reagent, 3,4-Dihydroxyphenylacetic acid (3,4-DHPAA). The assay was performed according to the manufacturer’s recommended protocol. We then determined the decreasing optical density of fibrous proteins with gelatin samples using the calibration curve of heparin. Gelatin concentrations with respect to heparin were 5 mg/mL, 10 mg/mL and 15 mg/mL. Finally, the amount of sGAGs in the L-ECM solution was quantified using the calibration curve of heparin standard.

### 4.9. Statistical Analysis

All of the values are presented as mean ± standard deviation (SD). Statistical comparisons were performed using a two-tailed Student’s t-test or one-way analysis of variance (ANOVA). P-values < 0.05 were considered to indicate statistically significant and P-values < 0.01 were considered a highly significant difference. 

## Figures and Tables

**Figure 1 jfb-10-00019-f001:**
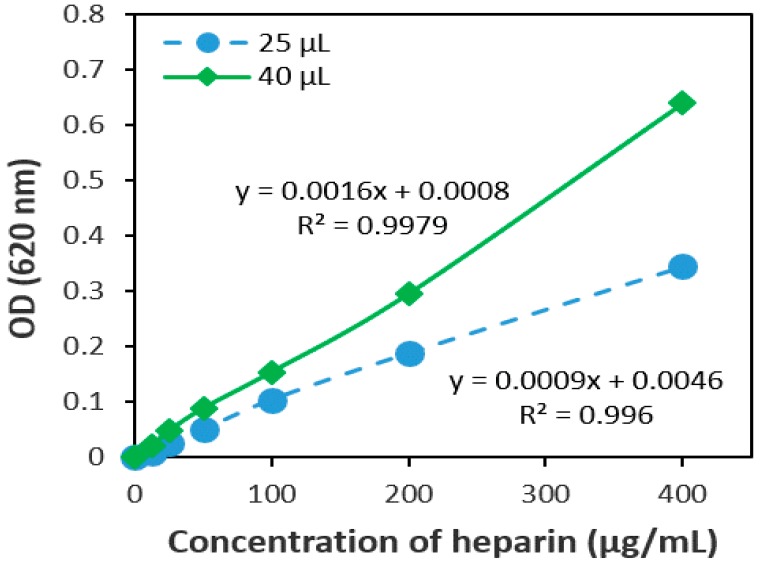
Linearity curves for 25 µL and 40 µL heparin samples mixed with the same volume of alcian blue (AB). The relationship between optical density and concentration was studied during the colorimetric reaction under experimental conditions. The R squared value was calculated, and the values depicted represent the mean (n = 3).

**Figure 2 jfb-10-00019-f002:**
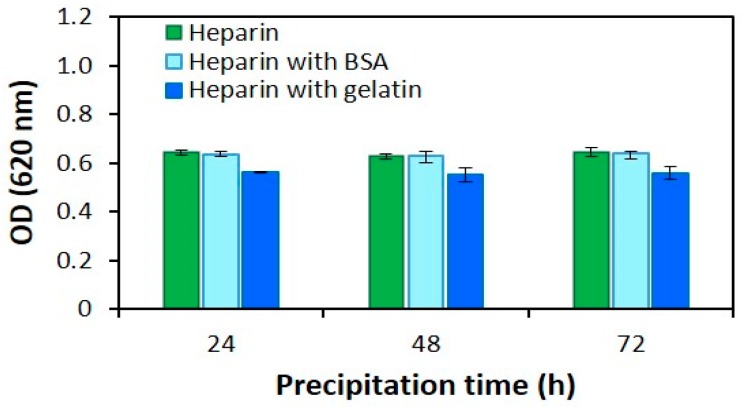
The optical density of heparin sample (400 µg/mL) at different time points (24 h, 48 h, and 72 h) during the precipitation process of heparin-AB. The concentration of proteins was 5 mg/mL. The values given represent the means ± SD and mean (n = 3).

**Figure 3 jfb-10-00019-f003:**
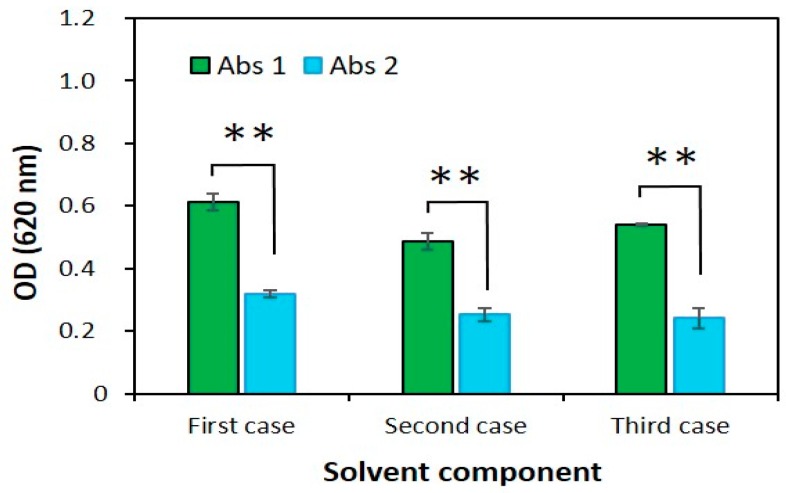
The optical density of the heparin sample (400 µg/mL) in different types of solvent used to dissolve the precipitate. The first case contains 5 N NaOH and 2-Aminoethanol, 1:49; the second case contains 5 N NaOH and 2-Aminoethanol, 1:4; and the third case contains 25 N NaOH and 2-Aminoethanol, 1:24. The data are shown as mean ± SD N = 3 and ** denotes where P < 0.01.

**Figure 4 jfb-10-00019-f004:**
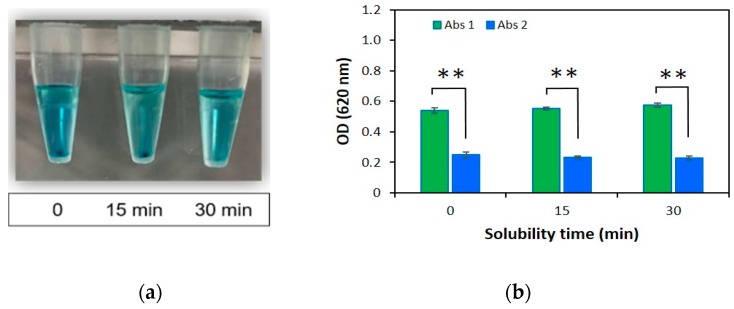
(**a**) Appearance of the samples in solubility time study, and (**b**) The optical density of heparin sample (400 µg/mL) at different time points (0 min, 15 min, and 30 min) before and after dilution with the third solvent component, respectively. The data are shown as mean ± SD N = 3 and ** P < 0.01.

**Figure 5 jfb-10-00019-f005:**
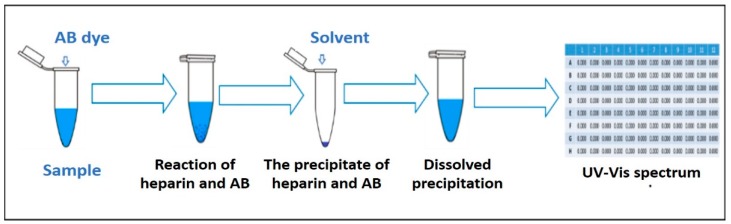
General schema of the AB method.

**Figure 6 jfb-10-00019-f006:**
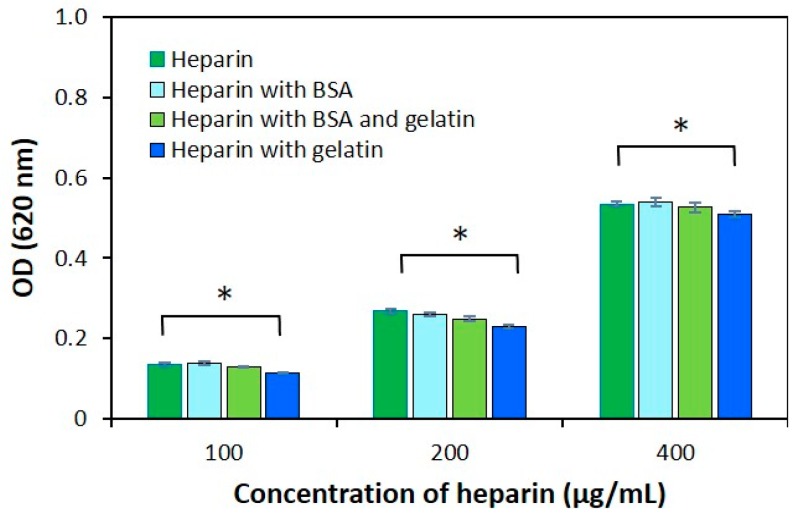
Comparison of the optical density of samples at different concentrations (100 µg/mL, 200 µg/mL and 400 µg/mL) of heparin alone, heparin with bovine serum albumin, and heparin with gelatin. The concentration of proteins was 5 mg/mL. The data are shown as mean ± SD N = 3 and * P < 0.05.

**Figure 7 jfb-10-00019-f007:**
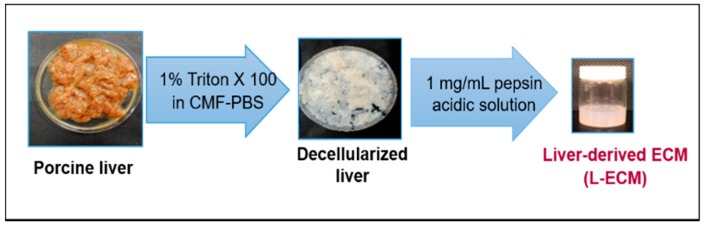
Design of Liver-derived ECM hydrogel preparation.

**Figure 8 jfb-10-00019-f008:**
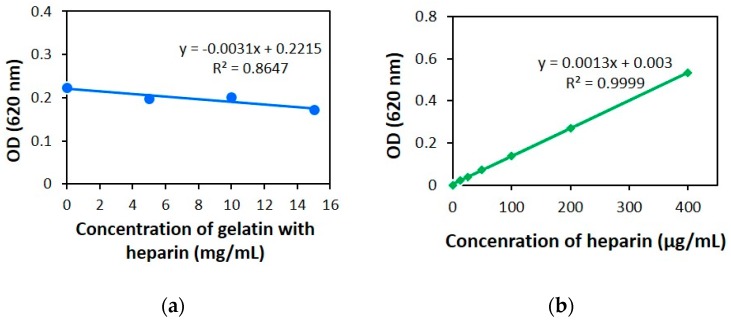
Calibration curves for (**a**) 100 µg/mL of heparin with 5 mg/mL, 10 mg/mL and 15 mg/mL of gelatin sample, and (**b**) Results were obtained from the concentration range of 12.5–400 µg/mL used for the heparin standard. The R squared value was calculated, and the values depicted represent the mean (n = 3).

**Table 1 jfb-10-00019-t001:** The dilution ratio of different types of solvent components.

Case No.	Chemicals	Ratio	Final Concentration of NaOH	The Dilution Ratio (OD.1/OD.2)
I	5 N NaOH and 2-Aminoethanol	1: 49	0.1 N	1.9
II	5 N NaOH and 2-Aminoethanol	1: 4.0	1.0 N	1.9
III	25 N NaOH and 2-Aminoethanol	1: 24	1.0 N	2.2

**Table 2 jfb-10-00019-t002:** The concentration and optical density for linearity study.

Concentration of Heparin (µg/mL)	400	200	100	50	25	12.5
Mean OD ^1^ (n = 3)	0.534	0.268	0.136	0.071	0.036	0.021
Standard deviation, SD	0.017	0.007	0.006	0.004	0.002	0.002

^1^ OD is the optical density of heparin at different concentrations

**Table 3 jfb-10-00019-t003:** Quantitative parameters of the heparin calibration curve.

Parameter	Value
λ_max_ (nm)	620.0
Linearity range (µg/mL)	12.5–400
Slope (b)	0.0013
Intercept (a)	0.003
Correlation coefficient, R^2^ (n = 3)	>0.996
SE of intercept	0.0005
SD of intercept	0.0013
Limit of detection (µg/mL)	2.95
Limit of quantification (µg/mL)	9.82
Recovery (%)	99.07

**Table 4 jfb-10-00019-t004:** Accuracy data from three different concentrations of heparin (n = 9).

Nominal Concentration of Heparin (µg/mL)	Recovery (%)			
	Case 1	Case 2	Case 3	Average
**25**	100.83	100.97	101.44	101.08
**50**	98.77	94.97	100.72	98.63
**100**	97.67	99.95	98.36	99.09

**Table 5 jfb-10-00019-t005:** Precision and accuracy performance data of reference heparin at different concentration levels.

**Intraday Precision**					
Nominal concentration (µg/mL)	Spiked concentration ± SD ^1^ mean value, (µg/mL)			CV^2^ (%)	RE^3^ (%)
	No. 1	No. 2	No. 3		
25	24.13 ± 0.89	24.87 ± 1.18	25.62 ± 1.54	4.80	0.515
50	51.82 ± 2.47	50.77 ± 2.31	47.92 ± 2.31	4.71	−0.341
100	97.46 ± 2.04	101.28 ± 1.94	102.79 ± 2.47	2.14	−0.51
**Interday precision**					
Nominal concentration (µg/mL)	Spiked concentration ± SD ^1^ mean value, (µg/mL)			CV^2^ (%)	RE^3^ (%)
	Day 1	Day 2	Day 3		
25	24.87 ± 1.18	25.18 ± 2.47	24.59 ± 1.60	7.02	0.48
50	50.77 ± 2.31	47.49 ± 2.70	49.97 ± 4.37	6.33	1.19
100	101.28 ± 1.94	98.77 ± 2.31	97.67 ± 5.12	3.16	0.77

^1^ Standard deviation, ^2^ coefficient of variation, and ^3^ relative error.

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
