# Peer review of "Analysis of Sulfated Glycosaminoglycans in ECM Scaffolds for Tissue Engineering Applications: Modified Alcian Blue Method Development and Validation"

_jfb, 2019, doi:10.3390/jfb10020019_

Round 1
Reviewer 1 Report
The manuscript reflects a carefully planned research and is documented by a comprehensive bibliography. It has a couple minor flaws:
- The study was carried out exclusively on heparin, which "has the highest negative charge density of any known biological macromolecule" (line 58). Please explain if and why the results obtained with heparin should apply also to others, less polar GAGs.
- at line 233, "It was reported that AB is insoluble in water..." while of course it is; the authors should explain more clearly what they mean in this paragraph.
- The division of proteins in "globular" and "fibrous", speaking of their interaction with sGAG, is too simplicistic and the topic should be discussed more extensively. For instance, sGAG are capable of a specific, D-periodic binding with collagen fibrils even in absence of protein cores (J Struct Biol 164:134-139, 2008), an interaction too specific to be due only to the collagen being fibrous.
Author Response
Response to Reviewer 1 comments
We would like to thank the editor and reviewers for their comments that helped us to improve our manuscript. The revision of this manuscript was based on reviewers’ comments and our responses are written in red as follows.
Reviewer 1
The manuscript reflects a carefully planned research and is documented by a comprehensive biography. It has a couple of minor flaws:
Point 1: The study was carried out exclusively on heparin, which “has the highest negative charge density of any known biological macromolecule”. Please explain if and why the results obtained with heparin should apply also to others, less polar GAGs.
Response 1: Thank you for bringing out this concern. Based on this comment and Reviewer 2’s suggestion, we focused on heparin qualification instead of all sGAGs in the L-ECM. Specifically, according to Reviewer 2’s suggestion, we have changed the title, modified a few statements in the introduction section and discussion points to focus more on heparin. As such, we added a more detailed explanation about heparin with corresponding references. Please see the line below.
lines: 52-56,
Heparin has the highest negative charge density of any known biological macromolecule whereas at least 70-80% of heparin is composed of the disaccharides, and the molecular weight distribution of heparin corresponds to its polydispersity. This structural variability makes heparin an extremely challenging molecule to characterize, and more sulfated, hence, more charged than other sGAGs. Furthermore, it is one of the well-known molecules used for tissue regeneration applications (An ACAD Bras CIenc. 2009, 81, 409-29.; J R Soc Interface 2015, 12 (110), 0589.; Cell Tissue Res 2010. 339. 237-46).
Also lines: 66-69,
Heparin is widely used as polyelectrolytes in multilayer built-up and the ECM components of hydrogel materials for tissue engineering applications. Physicochemical studies identify that the thermodynamics and kinetics of protein-heparin binding show considerable sensitivity than other sGAGs (Biotechnol. Bioeng. 2003, 82, 578–89.; Med. Biol. Eng. Comput. 2000, 38, 211–18.; J R Soc Interface 2015, 12 (110), 0589).
Point 2: At line 233, “It was reported that AB is insoluble in water…” while of course, it is; the authors should explain more clearly what then mean in this paragraph.
Response 2: Thank you for this comment and we apologize for the unclear explanation regarding this statement. In the manuscript, we added a few phrases to focus on this point. Please see the lines below.
Lines: 244-248
It was reported that the heparin and AB precipitate is insoluble in water as the ionic bonding between sGAG and AB is a stable interaction (Bunseki Kagaku. 2003, 52, 259–63.; NIH Public Access. Open Glycosci, 2008, 1, 31–9). A tetravalent cation with a hydrophobic core with positive charges attached strongly to heparin, the ionic strength, and other presence affect the precipitate in solution (Chronicles Young Sci. 2011, 2, 21.)
Point 3: The division of proteins in “globular” and “fibrous”, speaking of their interaction with sGAG, is too simplistic and the topic should be discussed more extensively. For instance, sGAGs are capable of a specific. D-periodic binding with collagen fibrils even in absence of protein cores (J Struct Biol 164: 134-139, 2008), an interaction too specific to be due only to the collagen being fibrous.
Response 3: This is a good observation and thanks you for bringing an important reference as an example. We revised the discussion section to emphasize heparin-proteins interaction. The sentences can be found in lines 305-311. Also, we added the references below.
In addition, some studies mentioned that heparin structure has the capability of interacting with more than one protein and activity should be viewed at the level of their interactions with multiple proteins (J R Soc Interface 2015, 12 (110), 0589.; Cell Tissue Res 2010, 339, 237-46). sGAGs are capable of a specific D-periodic binding with collagen fibrils even in absence of protein cores, an interaction too specific to be due only to the collagen being fibrous (J Struc Biol 2008, 164 (1), 134-39) Therefore, it remains an open question as to whether all heparin sequences interact to some degree with proteins and whether synergy effects are more relevant (Biologicals. 2008, 36, 134- 41).

Reviewer 2 Report
This article describes a dye-based method for the estimation of heparin concentration in complex matrices. The sample is mixed with a solution of Alcian Blue, leading to precipitation of a heparin-dye aggregate. This aggregate is then redissolved in a suitable solvent, and the absorbance measured to indicate heparin concentration. The assay is simple, uses easily available reagents, and the validation data provided indicate that it is suitable for heparin concentrations down to about 3 micrograms/mL.
Unfortunately, the authors make an additional claim that is not secure: a considerable amount of extra work needs to be done before this test is shown to be suitable for measuring the concentration of sGAGs in ECM scaffolds. Heparin is not a normal constituent of the ECM; the sGAGs in the ECM are chondroitin sufates, including dermatan sulfate; also keratan sulfate, and heparan sulfate. All of these GAGs have lower degrees of sulfation than heparin. It is necessary to establish validated calibration curves for each of these GAG types, not just for heparin.
If the authors do not wish to perform the extra experimental work necessary to justify the claims and title of this paper, they could choose to change the title, Introduction and Discussion to make it clear that this is a quantitative heparin assay, not a general sGAGs assay.
The authors should also compare the performance of their method with others in the literature, using dyes such as toluidine blue, methylene blue, and the commercial dye Heparin Red.
Some minor problems:
In several places in the manuscript, including the abstract, the ratio between dye and heparin is given in v/v terms. This is uninformative, and should be given in w/w terms, i.e. 10 mg heparin /1 mg dye.
The terms ‘crystallization’ and ‘crystallization time’ are used to describe the formation of an insoluble complex between AB dye and heparin. Is there evidence that crystals are formed? Should this word be replaced by ‘precipitation’, and ‘precipitation time’?
In the 7th line of the abstract, ‘sGAG-AB formation’ should be replaced with ‘sGAG-AB complex’.
The authors should check all their statistical analyses with care. The values for standard deviation in Table 5 are not correct. Raw data for other measurements, such as those for Fig. 1 for example, are not available and so cannot be checked.
Author Response
Response to Reviewer 2 comments
We would like to thank the editor and reviewers for their comments that helped us to improve our manuscript. The revision of this manuscript was based on reviewers’ comments and our responses are written in red as follows.
Reviewer 2
The article describes a dye-based method for the estimation of heparin concentration in complex matrices. The sample is mixed with a solution of Alcian Blue, leading to precipitation of a heparin-dye aggregate. The aggregate is then redissolved in a suitable solvent, and the absorbance measured to indicate heparin concentration. The assay is simple, uses easily available reagents, and the validation data provided indicate that it is suitable for heparin concentrations down to about 3 micrograms/mL.
Unfortunately, the authors make an additional claim that is not secure: a considerable amount of extra work needs to be done before this test is shown to be suitable for measuring the concentration of sGAGs in ECM scaffolds. Heparin is not a normal constituent of the ECM; the sGAGs in the ECM are chondroitin sulfates, including dermatan sulfate; also keratin sulfate, and heparan sulfate. All of these GAGs have lower degrees of sulfation than heparin. It is necessary to establish validated calibration curves for each of these GAG types, not just for heparin.
If the authors do not wish to perform the extra experimental work necessary to justify the claims and title of this paper, they could choose to change the title, introduction, and discussion to make it clear that this a quantitative heparin assay, not a general sGAGs assay.
The authors should also compare the performance of their method with others in the literature, using dyes such as toluidine blue, methylene blue, and the commercial dye Heparin Red.
Response: We very appreciate your comments and thank you sincerely. According to your comments, we modified the title, introduction and discussion section to focus on the Heparin. Additionally, the comparison between Alcian blue and methylene blue/toluidine blue assays were also discussed.
Some minor problems:
Point 1: In several places in the manuscript, including the abstract, the ratio between dye and heparin is given in v/v terms. This is uninformative and should be given in w/w terms, i.e. 10 mg heparin/ 1 mg dye.
Response 1: Thank you for this valuable comment. We checked the concentration carefully and changed the unit to w/w.
Lines: 17 and 384
Point 2: The terms “crystallization” and “crystallization time” are used to describe the formation of an insoluble complex between AB dye and heparin. Is there evidence that crystals are formed? Should this word be replaced by “precipitation” and ”precipitation time”.
Response 2: We are glad that you highlighted such important detail. Indeed, we should define clear expressions to avoid confusion. In this regard, we modified all expressions in the text and figure as below:
à crystallization ---- precipitation;
à crystallization time --- precipitation time;
à crystal --- precipitate.
Please see lines 16, 113, 114, 118, 119, 123, 140, 145, 253 and Figures 2 and 5 in the manuscript.
Point 3: In the 7th line of the abstract “sGAG-AB formation” should be replaced with “sGAG-AB complex”.
Response 3: Thank you for this comment. We replaced the expression based on your suggestion. The modified statements can be found in lines 18, 25, 89, 116 and 149.
Point 4: The authors should check all their statistical analysis with the case. The values for the standard deviation in Table 5 are not correct. The raw data for other measurements, such as those for Fig.1 for example, are not available and so cannot be checked.
Response 4: Thank you for pointing this out and we clearly apologize for this mishap. Specifically, some miscalculations were made in Table 5 regarding the standard deviation. According to your advice, we checked the raw data again and performed the statistical analysis in all tables carefully and then calculated correctly. In the paper, we did not show the data for Table 5, and we are sorry again that information such as this was not supplied in the manuscript.
In the manuscript, we showed a revised version of Table 5 in the new format.

Round 2
Reviewer 2 Report
The authors have modified this manuscript with good intentions, but some of the changes have made the paper worse rather than better. This seems to be due to a misunderstanding. To repeat words from the earlier report: Heparin is not a normal constituent of the ECM; the sGAGs in the ECM are chondroitin sulfates, including dermatan sulfate; also keratan sulfate, and heparan sulfate.
Heparin is not a common sGAG in the ECM; it is only found in the secretory granules of mast cells. The authors are using heparin because is it easily available, and because it is used in the design of artificial scaffolds for tissue engineering, as a substitute for natural sGAGs. It is not sufficient to simply replace the word ‘sGAGs’ with ‘heparin’ throughout. This is particularly important for the determination of sGAG in liver ECM described in sections 2.4 and 3.4. The authors have not measured the amount of heparin in liver ECM; they have used a heparin standard curve to quantify sGAGs in liver ECM, and for this reason their measurement is inaccurate. As the authors clearly do not wish to delete this section, they need to be clearer about the inaccuracy they introduce in this way.
In more detail:
Title: Please revert to the earlier version.
Abstract, 1st sentence, lines 13-15: Either revert to the original sentence or delete this sentence.
Line 19: Please delete ‘in a 1:1 (w:w ) ratio’.
Lines 27-29: Replace this sentence with “Further, we measured the content of sGAGs (expressed as heparin equivalent) in the ECM from a decellularized porcine liver.”
Line 60: change to read “..at least 70-80% of heparin is composed of a trisulfated disaccharide,…”
Line 64: Please revert to the earlier version.
Line 83-86 , sentences beginning ‘In addition…’ and ‘These carboxyl groups…’: Please revert to the earlier versions.
Lines 97-99: Change to “After being developed, the AB method has been successfully applied for the determination of the sGAG amount, (expressed as heparin equivalent) in ECM scaffolds from the porcine liver.
Table 5: Please give the value of n for these results: is it 9? Why are some values for RE negative, as RE is a ratio expressed as %?
Line 236: Please revert to earlier version of the title to section 2.4.
Line 237: Please revert to the earlier version.
Line 249: Please replace ‘spectra’ with ‘dose-response lines’
Line 250: Please change to “The sGAG amount in L-ECM was 40 µg/ml, expressed as heparin equivalent”.
Lines 250-251: Please delete the sentence beginning “This result corroborated…..” It has no scientific basis.
Lines 258-259: Please revert to the earlier version.
Lines 346-348, sentence beginning “Theoretically,…” Please revert to the earlier version.
Lines 349-350, sentence beginning “ In both..” Please revert to the earlier version.
Lines 352-354, sentence beginning “OD of L-ECM…” Please revert to the earlier version.
Lines 361-363: Please change to “Finally, we derived the ECM from porcine liver and determined the amount of sGAGs contained in it considering the presence of proteins in the L-ECM, expressing the results in terms of heparin equivalents, against a heparin standard curve. As ECM sGAGs have substantially different structures to heparin, the value obtained is likely to be inaccurate in absolute terms.”
Lines 363-364: I suggest "Additionally, our finding suggested that this modified AB method may be applied to artificial, heparin-containing scaffolds and other wide-ranging applications in the tissue engineering field."
Line 454: Please revert to the earlier version of the title to section 4.8.
Lines 455 and 462: Please revert to the earlier versions.
Author Response
Response to Reviewer 2 comments
Thank you so much for your more detailed and valiant comments. We revised our manuscript (written in red) according to your comments.
The authors have modified this manuscript with good intentions, but some of the changes have made the paper worse rather than better. This seems to be due to a misunderstanding. The repeat words from the earlier report: Heparin is not a normal constituent of the ECM; the sGAGs in the ECM are chondroitin sulfate, including dermatan sulfate; also keratin sulfate, and heparan sulfate.
Heparin is not a common sGAG in the ECM; it is only found in the secretory granules of mast cells. The authors are using heparin because is it easily available, and because it is used in the design of artificial scaffolds for tissue engineering, as a substitute for natural sGAGs. It is not sufficient to simply replace the word ‘sGAGs’ with ‘heparin’ throughout. This is particularly important for the determination of sGAG in liver ECM described in sections 2.4 and 3.2 The authors have not measured the amount of heparin in liver ECM; they have used a heparin standard curve to quantify sGAG in liver ECM, and for this reason their measurement is inaccurate. As the authors clearly do not wish to delete this section, they need to be clearer about the inaccuracy they introduce in this way.
In more details:
Point 1: Title, Please revert to the earlier version.
Thank you for your advice. The title was changed as below.
Analysis of sulfated glycosaminoglycans in ECM scaffolds for tissue engineering application: Modified alcian blue method development and validation
Point 2: Abstract, 1st sentence, lines 13-15: Earlier revert to the original sentence or delete this sentence.
Thank you for your comment. We reverted to the original sentence.
Determination of the accurate amount of glycosaminoglycans (GAGs) in the complex mixture of extracellular matrix (ECM) proteins is important for tissue morphogenesis and homeostasis.
Point 3: Line 19, please delete ‘in a 1:1 (w/w) ratio’.
According to your this comment, we deleted the words from the sentence. The full sentence is here.
The AB dye and heparin sample were allowed to react at 40C for 24 hours.
Point 4: Lines: 28-30; replace this sentence with “Further, we measured the content of sGAGs (expressed as heparin equivalent) in the ECM from a decellularized porcine liver”.
Point 5: Line 61; change to read “… at least 70-80% of heparin is composed of a trisulfated disaccharide,..”
Thank you for your meaningful comments and we changed the sentences as below.
Lines 28-30. We changed the sentence as your advice.
Line 61: Heparin has the highest negative charge density of any known biological macromolecule whereas at least 70-80% of heparin is composed of a trisulfated disaccharide, and the molecular weight distribution of heparin corresponds to its polydispersity.
Point 6: Line 65; please revert to the earlier version.
Point 7: Line 84-86; the sentences beginning ‘In addition...’ and ‘These carboxyl groups…’: Please revert to the earlier versions.
Thank you for your meaningful comments and we converted the following sentences as below.
Line 64: sGAG play an important role in chemical signaling between cells by binding to and regulating the activity of various secreted signaling molecules and proteins
Line 84-86: In addition, sGAGs are covalently bonded to core proteins residing in the ECM as well as proteins containing a number of negatively charged carboxyl groups and a hydrophobic domain.
Line 86-87: These carboxyl groups in a protein can be eliminated using the AB method at a low pH, which might hinder the electrostatic interaction between sGAG and AB.
Point 8: Line 97-99; Change to “After being developed, the AB method has been successfully applied for the determination of the sGAG amount (expressed as heparin equivalent) in ECM scaffolds from the porcine liver”.
Line 97-99: Thank you so much again, this sentence was changed as mentioned above.
Point 9: Table 5; please give the value of for these result: Is it nine? Why are some values for RE negative, as RE is a ratio expressed as %?
We have been used this formula with the value of RE;
(1) | |
The studies were used 25 µg/mL, 50 µg/mL and 100 µg/mL of heparin solution. Then we calculated the RE of all case by equation 5.
For example: In the intraday precision study, for 100 µg/mL of heparin solution, the nominal concentration equals 100.
From the results, we obtained the spiked concentration.
No. | Spiked concentration |
1 | 99.77 |
2 | 95.92 |
3 | 96.69 |
4 | 103.08 |
5 | 99.23 |
6 | 101.54 |
7 | 101.77 |
8 | 105.62 |
9 | 101.00 |
Average | 100.51 |
The relative error was calculated:
Point 10: Line 238; please revert to the earlier version of the title to section 2.4.
Point 11: Line 239; please revert to the earlier version
We reverted all points with your comments. Thank you.
Line 238: 2.4. Quantification of sGAGs in the L-ECM
Line 239: An optimized AB method was used to determine the accurate amount of sGAGs in native tissue L-ECM derived from decellularized porcine liver.
Point 12: Line 250; please replace ‘spectra’ with dose-response lines’.
Thank you for the comment. The sentence was changed as below.
Line 250: The dose-response lines of the heparin with the serial concentration are presented in Figure 8 b.
Point 13: Line 250; Please change to “The sGAG amount in L-ECM was 40 µg/ml, expressed as heparin equivalent”.
Point 14: Line 252-253; Please delete the sentence beginning “This result corroborated …”. It has no scientific basis.
Line 250: We changed this sentence as your advice.
Line 252-253: The sentence was deleted. We apologize that the sentence was the unscientific principle.
Point 15: Line 260-261; please revert to the earlier version
Point 16: Line 350-352; the sentence beginning “Theoretically…”Please revert to the earlier version.
Point 17: Line 352-354; the sentence beginning “In both…”Please revert to the earlier version.
Point 18: Line 356-358; the sentence beginning “OD of L-ECM…”Please revert to the earlier version.
Thank you for all the comments. The sentences were converted the earlier versions as below.
Line 260- 261; AB has been one of the most widely used cationic dye for acidic polysaccharides such as sGAGs since 1960.
Line 350-352: Theoretically, hundreds of proteins interact with sGAGs, which are directly involved in various signaling pathways.
Line 352-354: In both in vitro and in vivo studies, researchers have been able to detect the exact amount of sGAG.
Line 356-358: OD of L-ECM was compared to a standard based on heparin, then total sGAG content was expressed as µg/mL of heparin
Point 19: Lines 365-369; please change to “Finally, we derived the ECM from the porcine liver and determined the amount of sGAGs contained in it considering the presence of proteins in the L-ECM, expressing the results in terms of heparin equivalents, against a heparin standard curve. As ECM sGAGs have substantially different structures to heparin, to value obtained is likely to be inaccurate in absolute terms.
Point 20: Lines 369- 371; I suggest “Additionally, our finding suggested that this modified AB method may be applied to artificial, heparin-containing scaffolds and other wide-ranging applications in the tissue engineering field”.
We are glad for your great help.
Lines 365-369 and 369- 371: We fully changed according to your advice.
Point 21: Line 460; please revert to the earlier version of the title to section 4.8.
Point 22: Lines 461 and 468; please revert to the earlier versions.
Thank you meaningful comments, we changed those sentences..
Line 460: The title was changed as below.
4.8. The amount of sGAG in solubilized L-ECM
Line 461:
To investigate the efficacy of the modified AB method, the amount of sGAGs in solubilized L-ECM was determined with reference to the fibrous protein content.
Line 468:
Finally, the amount of sGAGs in L-ECM solution was quantified using the calibration curve of heparin standard.

Round 3
Reviewer 2 Report
The authors have addressed all comments, and the paper is now OK.